# Air Pollution, Environmental Protection Tax and Well-Being

**DOI:** 10.3390/ijerph20032599

**Published:** 2023-01-31

**Authors:** Jingjing Wang, Decai Tang

**Affiliations:** 1School of Law and Business, Sanjiang University, Nanjing 210012, China; 2Panyapiwat Institute of Management, Bangkok 11120, Thailand; 3School of Management Science and Engineering, Nanjing University of Information Science & Technology, Nanjing 210044, China

**Keywords:** air pollution, environmental protection tax, subjective well-being, mediation effect, instrumental variable

## Abstract

The effective control of air pollution to advance human health and improve well-being has risen to the forefront of discussion in recent years. Based on China’s 2019 environmental protection tax data and China Social Survey (CSS) data, this paper studies the effects of subjective air pollution and the environmental protection tax on residents’ well-being using an econometric mediation effect model. The research conclusions are as follows: (1) Subjective air pollution can significantly reduce residents’ well-being, (2) an environmental protection tax can significantly improve residents’ well-being and it can eliminate some of the negative influence of subjective air pollution on residents’ well-being, and (3) the grouping test of residents’ income, regional distribution, urban and rural structure, age structure, gender structure, and other variables shows that the effects of subjective air pollution on residents’ well-being are heterogeneous among different populations. After further endogeneity testing with the instrumental variables method, adjusting the primary variables, and altering the research procedures, the results are still robust. Based on these findings, we should vigorously promote the development of ecological civilization and good air quality and support reforming the environmental protection tax system to enhance well-being. It is also necessary to shift from a crude development model to a green industry and business model. While emphasizing social equity and production efficiency, we should ensure the synchronous development of cities and villages. Additionally, tangible steps should be implemented to raise people’s incomes, expand young people’s work options, and enhance their satisfaction. The article focuses on the impact of subjective air pollution on residents’ well-being, adding air pollution to the factors affecting well-being. Furthermore, the article finds that the environmental protection tax has two advantages: it can govern air pollution and promote green development, and, at the same time, it can enhance social harmony and improve residents’ well-being.

## 1. Introduction

Since opening to the outside world, China’s economy and society have developed rapidly. In recent years, China’s GDP has grown rapidly, and the living standard of residents has improved significantly. However, environmental problems have also come along; water, soil, air, and other environmental pollution are increasingly becoming serious. Compared with other pollution phenomena, the effect of air pollution on the happiness of Chinese residents is more obvious. Residents have more direct contact with regional air pollution, whether at the visual, olfactory, or psychological level, and it has been shown that air pollution endangers people’s physical health to some extent [1,2,3,4] and may reduce their cognitive ability [5]. Air pollution also increases economic and social costs and restricts the sustainable development of the economy and society [6,7]. In addition, air pollution impacts people’s well-being; it has a negative impact on people’s living environment and brings harm to their health, which likely reduces residents’ well-being [8,9]. The sacrifice of the ecological environment for economic growth has reduced residents’ sense of happiness. However, economic growth has also improved residents’ material living standards and their sense of happiness. 

According to the Gallup World Poll, more than 80% of those polled chose to give priority to protecting the ecological environment rather than accelerating economic development [10]. For a long time, China lacked a true environmental protection tax system. Instead, the environmental protection tax system was replaced by an emission fee system. The Environmental Protection Tax Law was signed into law on 25 December 2016 and went into effect on 1 January 2018 [11]. This marked the end of the emission fee system and the beginning of the environmental protection tax system. China then began to use fiscal and tax policies, as well as legal means, to control environmental pollution [11]. The environmental protection tax certainly increases the cost of enterprises while considering the environment [12], but how does it ultimately impact people’s well-being? Does the environmental protection tax play a mediating role between air pollution and people’s well-being? Do people in different regions, income levels, and urban and rural areas differ in their well-being between air pollution and the environmental protection tax?

To sum up, this paper uses China Social Survey (CSS) data and environmental protection tax data to make a correlation analysis of Chinese residents’ well-being to (1) study the effects of subjective air pollution on residents’ well-being; (2) study the effects of environmental protection tax on residents’ well-being; and (3) study whether different people’s subjective air pollution has different effects on happiness.

## 2. Literature Review and Hypotheses

### 2.1. Subjective Air Pollution and Residents’ Well-Being

One of the biggest health risks is environmental pollution, which includes noise, air, and water contamination. According to the World Health Organization, environmental factors are responsible for around 25% of diseases worldwide [13]. Ferreira’s team took European residents as research objects to study the impact of air pollution on happiness. The results show that sulfur dioxide can reduce residents’ happiness [14]. Zang et al. found that the negative impact of air pollution on the well-being of Chinese residents is different due to the impact on residents’ physical health and well-being [15]. Ahumada and Iturra found that air pollution reduced residents’ well-being based on the air particle data of Chilean cities in 2013 [16]. In addition to studying the harm that objective air pollution brings to the body and well-being, a few scholars have also studied the effects of subjective air pollution on well-being. The air pollution value of residents’ subjective perception and the actual measured air pollution value reduce residents’ life satisfaction [17]. Song’s research found that residents’ subjective perception of air pollution reduced their happiness [18]. Ye et al. also arrived at the same result [19]. The well-being of people will surely increase as the environmental quality is improved because environmental contamination has been found to lead to bodily and mental disorders as well as decreased well-being. Additionally, improvements in air quality are favorably correlated with well-being, according to Sanduijav et al. [20].

None of the aforementioned research used data from China’s 30 provinces and autonomous regions in 2019, despite the fact that they all hold that air pollution can lower people’s well-being. Furthermore, despite the fact that subjective air pollution and well-being are correlated, the endogenous problem is rarely taken into account in these investigations.

### 2.2. Research on Air Pollution and Environmental Protection Tax

To control air pollution, there is a variety of legal, administrative, financial, and tax options. Environmental taxation system creation and improvement are now crucial components of reducing environmental pollution. Environmental protection taxes are effective at preserving the environment and lowering pollutant emissions. Countries are increasingly using it as a key tool to tackle environmental pollution. Pearce proposed that environmental taxes have two major functions: improving environmental quality and raising taxes [21]. According to Bovenberg, an environmental protection tax is a neutral tax that has the ability to increase employment, lower tax costs, and improve environmental quality [22]. Deng Liping and Chen Bin studied China’s environmental protection tax system and found that China’s environmental tax reform can significantly reduce air pollution [23], while Kwilinski et al. believe that the environmental tax is an effective measure to improve the quality of the regional ecological environment. The environmental tax has positive effects on pollutant emission reductions [24]. The above-mentioned literature considers environmental taxation as an important tool to combat environmental pollution, including air pollution, so the studies mainly focus on the establishment and improvement of the environmental protection tax system and the effects of policy implementation. Some scholars have also focused on the causal relationship and quantitative relationship between subjective air quality and environmental taxation, though no studies have been found to give clear conclusions on the quantitative relationship between the subjective air quality and environmental taxation.

### 2.3. Research on Environmental Protection Tax and Well-Being

Taxes for environmental protection help strike a balance between ecological sustainability and economic growth. Environmental taxation can be used for urban construction and environmental pollution control to promote regional sustainable development. There have not been a lot of scholarly studies on how the tax code, tax rate, and overall tax burden affect residents’ well-being. However, a global Gallup survey was used by Oishi et al. to demonstrate that progressive tax is favorably correlated with well-being [25] and according to Drus M.‘s research findings, taxes have a favorable, significant, and strong impact on well-being [26]. Akay et al.’s research also show that taxes have a noticeably favorable effect on people’s well-being [27], i.e., the government can boost spending on public services such as education and transportation while also enhancing its own well-being [28,29].

The findings of China’s studies on taxes and well-being are neither identical nor incompatible. The research of Tang and Su found that the growth value of environmental tax has a promoting effect on the subjective well-being of Chinese residents [30] while Xie Shun et al. believe that the macro tax burden has a significant negative impact on well-being [31]. Some scholars have also found that China’s taxation does not have a significant impact on happiness; there is an inverted “U” relationship between Chinese tax revenue and residents’ well-being [32]. In addition to the aforementioned research, some researchers point out that the macro tax burden and well-being do not necessarily have a negative (positive) relationship [33]. All the above studies have studied well-being from the perspective of the overall tax burden, and no studies have been found to reveal the relationship between environmental tax and happiness.

In summary, studies on air pollution on well-being mainly focus on objective air pollution, such as CO_2_, SO_2_, PM2.5, etc. In contrast, there are relatively few studies on subjective air pollution and well-being. Studies on the impact of tax on well-being have not yet reached a consistent conclusion. There is a paucity of research on environmental tax and well-being. Wang et al. have confirmed that green tax, including environmental protection tax, significantly enhances people’s well-being [34]. However, no research has been conducted on the effects of the environmental protection tax on well-being. In order to improve the research system, this paper adds environmental protection tax variables to further analyze the impact of the environmental protection tax on residents’ well-being. It further provides a novel perspective for a deeper understanding of the intricate factors behind the impact on well-being. It is intended to enrich the research on well-being from the perspective of subjective air pollution and, simultaneously, to explore the impact of the environmental protection tax on well-being in a pioneering way.

The following is the structure of this paper’s organization: First, the research background and existing research results of residents’ well-being are sorted out. Secondly, the impact mechanism of subjective air pollution and the environmental protection tax on residents’ well-being is analyzed. Then, the research hypotheses of this paper are put forward. The description of the model parameters and data sources are next given, followed by a description of the metrological tests and their findings. Finally, the limits and suggestions for future research are explored together with the research findings and policy recommendations.

## 3. Influence Mechanism and Research Hypothesis

### 3.1. Influence Mechanism Analysis

When the environmental protection tax paid by enterprises is higher than the pollution governance cost, enterprises actively choose to increase their investment in pollution governance and energy efficiency so as to realize the upgrading of production technology and reduce the payment of the environmental protection tax [35]. In general, a tax on environmental protection mainly influences how prices change, which may have an impact on social behavior and result in a “green” shift. Equally important is encouraging the preservation of the ecosystem and its resources. The environmental protection tax can not only effectively reduce the pollution emissions of enterprises, but also protect the ecological environment. It can also guide enterprises to upgrade technology, improve production levels, increase enterprise income, and promote sound and rapid economic development. Therefore, the environmental protection tax plays an important role in promoting environmental pollution control, economic growth, and social sustainable development [36,37]. Figure 1 depicts the link and mechanism of influence between air pollution, the environmental protection tax, and well-being.

### 3.2. Research Hypotheses

Smog and sandstorms are frequent in China, and the country has a major air pollution problem that needs to be handled. Air pollution lowers household income, stunts economic growth, and causes significant economic losses to society. Studies show that a person’s gender, age, marital status, income, and so on all have an impact on their well-being [38]. Based on an economic and psychological analysis, air pollution not only affects residents’ health (such as the respiratory system and cardiovascular and cerebrovascular diseases) but also causes residents to suffer from psychological diseases such as depression, thus leading to the decline of residents’ well-being. Taxes on environmental protection are a crucial tool in the battle against air pollution. An environmental tax will compel firms to modernize their industries and adopt green innovations, which will ultimately lead to the harmonious coexistence of nature, businesses, and people as well as lead to an improvement in social welfare and well-being. As a result, the following hypotheses are put forth in this paper:

**H1:** 
*Well-being can be lowered by air pollution.*


**H2:** 
*The amount of money that the environmental protection tax can generate increases with air pollution.*


**H3:** 
*The tax on environmental protection can raise people’s subjective well-being.*


**H4:** 
*The environmental protection tax mediates well-being and air pollution.*


Uneven regional growth, an absence of coordination between urban and rural development, and the rising wealth inequality are the principal manifestations of China’s persistent challenge of uneven and insufficient economic and social development. As a result, individual variances among residents are very prevalent. Influenced by urban and rural structure, residents’ income level, age structure, and other factors, the influence of subjective air pollution on people’s lives is also different. Therefore, the subjective air pollution assessment of different residents is also different. To increase their sense of well-being, high-income groups favor green consumption. There is a positive correlation between residents’ demand for a good air environment and their income level. Low-income residents give priority to material consumption, i.e., their sense of well-being is improved by improving their material life level [39]. Moreover, the effects of subjective air pollution and residents’ well-being may vary according to gender and age groups. The fifth hypothesis is thus proposed based on the above:

**H5:** 
*The effects of subjective air pollution on residents’ well-being are heterogeneous depending on gender, age, income, region, and urban/rural areas.*


## 4. Research Design

### 4.1. Data Sources

The 2019 Chinese Social Survey (CSS) is the data source for this paper. The survey was started in 2003 and is conducted annually by the Renmin University of China. The survey primarily focuses on the living, economic, and working conditions, social security, social values, and social evaluation of China cities and village families, containing more than 11.6 million data items [40]. The survey is conducted in more than 11,000 China cities and village families, covering 30 provinces and 596 villages. The China Statistical Yearbook provides information on environmental protection tax.

### 4.2. Empirical Model

By using Levinson’s econometric model [41], this research develops a well-being model made up of air pollution, environmental protection tax, and well-being to investigate the effects of these factors on well-being.
(1)$${\mathrm{SWB}}_{ij}=\beta_{0}+\beta_{1}{\mathrm{airpol}}_{ij}+{\;\beta }_{2}{\mathrm{envt}}_{j}+Z_{ij}+{\;\varepsilon }_{ij}$$
where ${\mathrm{SWB}}_{ij}$ is the well-being of resident *i* in province *j*; ${\mathrm{airpol}}_{ij}$ is the perceived level of air pollution by resident *i* in province *j*; ${\mathrm{envt}}_{j}$ is the environmental protection tax in province *j*; $Z_{ij}$ includes the characteristics of resident *i* in province *j*; $\varepsilon_{ij}$ is the term of random disturbance; $\beta_{0}$ is a constant term; $\beta_{1}$ is the regression coefficient of air pollution on well-being; and $\beta_{2}$ measures the degree of the environmental protection tax on well-being. The symbols indicate the impact’s direction.

### 4.3. Variable Selection

This paper’s explained variable is well-being (SWB), with air pollution as the core explanatory variable. The environmental protection tax is the mediating variable. The fundamental control variables found in the majority of studies on well-being are included in this paper’s model: age, marriage, education, income, and other factors [42]. In addition, we add urban and rural structure and regional structure variables to further analyze the influence of these two variables on residents’ well-being [43]. Table 1 displays all variables’ descriptive statistics.

To ensure the reliability of the research conclusions, the data used in this paper are described as follows: (1) Since the CSS data in 2019 does not include the data from Xinjiang, Hong Kong, Macao, and Taiwan, this paper focused on 30 provinces. (2) The data of control variables were taken from Part A, and the total number of samples was 10,286. (3) Residents’ well-being was selected from CAPI, with more than 5000 samples. (4) The subjective air pollution data are from Part G, and the total number of samples was 10,286. To ensure data unification, 5112 sample data were used. (5) Invalid data with unclear expressions were deleted from the questionnaire, and finally, 4837 valid sample data were used in this paper.

## 5. Empirical Test

### 5.1. Basic Regression Results

To avoid the deviation of the regression results, this paper conducts a multicollinearity test on the research variables. One method to check for multicollinearity is to use the variance inflation factor (VIF). The average VIF is 1.25 after removing the variable age squared. All VIFs are below 2, demonstrating that the model has no multicollinearity problems [44].

Both ordered probit and OLS estimation results show consistency in symbols and results, as numerous researchers have demonstrated [45]. Because the mediating effect between variables is to be tested, the traditional OLS method model is used for regression in this paper and a robustness test is carried out. The regression results of air pollution and environmental protection tax on well-being are shown in Table 2.

The regression results showed that the subjective air pollution significantly reduced residents’ happiness. If the subjective air pollution value increases by 1, residents’ happiness will decrease by 0.0351. Subjective air pollution reduces residents’ happiness. Therefore, assumption 1 in this paper is correct. The regression results of model (2) show that the subjective air pollution has a positive promoting effect on the environmental protection tax. The environmental protection tax will rise by 1.133 units for each additional unit of air pollution, supporting hypothesis 2. The regression results of model (3) show that the environmental protection tax can improve residents’ happiness. Therefore, assumption 3 is also true. Well-being is significantly correlated with education, income, age, age squared, and marital status among the control variables.

### 5.2. Mediation Effect Mechanism Test

According to models (2) and (3) in Table 2, there is a significant direct effect with a coefficient of −0.036 between well-being and air pollution, i.e., c’ = −0.036, which means that air pollution significantly reduces residents’ well-being. The indirect effect ab is also significant, with a coefficient of 0.000761. The results show that the environmental protection tax can improve residents’ happiness. According to Wen and Ye [46], the environmental tax plays a “covering effect” rather than a mediating effect between subjective air pollution and residents’ well-being, which shows that hypothesis 4 is not true. The reduction effect of some subjective air pollution on residents’ happiness is “covered” by environmental taxes because it improves residents’ happiness. The ratio of covering effect to direct effect |ab/c| = 2.11% can also be calculated. The overall, direct, and masking effects of subjective air pollution on residents’ happiness are shown in Figure 2.

### 5.3. Heterogeneity Research

Air pollution has been found to have a significant negative impact on well-being in previous studies. We will continue to research the gender, age, and income heterogeneity of this conclusion using the CSS2019 data.

#### 5.3.1. Income Heterogeneity Test

Table 3 depicts the effects of subjective air pollution on the well-being of various income groups. The high-income group consists of individuals whose annual income is higher than the average, while the low-income group consists of individuals whose annual income is lower than the average.

According to Table 3, subjective air pollution reduces well-being in both high-income and low-income groups, however, the degree is different. One unit of air pollution causes a 0.0226 unit decrease in the well-being of the high-income group and significantly causes a 0.0402 unit decrease in the low-income group. The reduction in well-being for the low-income group is 1.78 times greater than that for the high-income group. The high-income groups have a higher ability to circumvent air pollution than the low-income groups due to their better economic base. Usually, the high-income groups work mostly in indoor environments, while the low-income groups are often exposed to outdoor work. The low-income groups are constrained by economic conditions and have to trade off between air quality and economic costs, thus suffering greater welfare losses from air pollution [47].

#### 5.3.2. Region Heterogeneity Test

China’s geographical space is usually divided into the eastern region, the western region, and the central region. The central and western regions are grouped as the central and western region in this paper. Table 4 displays the results of the region regression.

Table 4 shows that subjective air pollution in both the eastern and central and western regions of China brings about a decrease in well-being. Although it is not significant in the eastern region, 1 unit of air pollution causes a 0.0237 unit decrease in well-being in the eastern region whereas 1 unit of air pollution causes a significant 0.057 unit decrease in well-being in the central and western region. The decrease in well-being in the central and western region is 2.41 times greater than in the east. Based on the “optimal allocation of resources” production activities, the eastern region is more developed. In contrast, the central and western region is relatively backward. In addition, the eastern region consumes more resources and environment than the central and western region by pursuing lower factor costs and moving their highly polluting and energy-consuming industries to less developed regions, thus making the consequences of air pollution mainly borne by the less developed central and western region. People in the central and western region have to endure the reduced well-being brought about by air pollution to obtain more income and employment opportunities.

#### 5.3.3. Urban-Rural Heterogeneity Test

Residents are divided into urban and rural areas in this paper. The heterogeneity of the effect of air pollution on well-being between urban and rural groups is shown in Table 5.

Table 5 shows that subjective air pollution causes a decrease in well-being for both urban and rural residents in China. Although it is not significant, 1 unit of air pollution reduces urban inhabitants’ well-being by 0.0202 units. For rural inhabitants, 1 unit of air pollution significantly reduces well-being by 0.0413 units, therefore, rural residents experience a 2.04 times larger decline in well-being than urban residents. The Chinese government has implemented strict environmental control policies in recent years and the “Blue Sky Defense” plan for key cities and regions, which has resulted in significant improvements in air pollution in these cities and regions [48]. As a result, many polluting industries and sectors have shifted to rural areas, which has increased air pollution and reduced the well-being of rural residents.

#### 5.3.4. Age Heterogeneity Test

This paper examines the heterogeneity of the effect of subjective air pollution on well-being across age groups, using 45 years as the cut-off, defining those above 45 years as elderly people and those below 45 years as young people. The regression results are shown in Table 6.

Table 6 shows that air pollution causes a decrease in well-being for both the elderly and young people. Although it is not significant for the elderly, 1 unit of air pollution causes a decrease of 0.0318 units of well-being for the elderly while 1 unit of air pollution causes a significant decrease of 0.0372 units of well-being for the young. The decline in well-being among young people is 1.17 times greater than that of older people. Compared to the elderly, the young engage in more outdoor activities/work and suffer more from air pollution. The elderly, on the other hand, have a choice of staying indoors when air pollution is serious or choosing to move to an area with good air quality.

#### 5.3.5. Gender Heterogeneity Test

Table 7 shows the gender heterogeneity test result.

Table 7 shows that air pollution brings about a decrease in well-being in both males and females. Although not significant for females, 1 unit of air pollution causes a 0.0187 unit decrease in well-being, whereas 1 unit of air pollution causes a significant 0.0592 unit decrease in well-being for males. Therefore, men’s well-being decreases 3.16 times more than women’s. Men engage in more work and outdoor activities than women and are more sensitive to the perception of subjective air pollution, whereas women typically work and engage in activities indoors; thus, the effects of subjective air pollution on their happiness are not significant.

### 5.4. Robustness Check

For robustness testing, various statistical approaches or substituting of core variables are typical. By substituting core variables, the robustness test is carried out in this paper; life satisfaction has taken the place of well-being. As shown in Table 8, the subjective air pollution will reduce residents’ happiness. The environmental protection tax still plays a covering effect between air pollution and life satisfaction, with a covering effect value of 2.27%. Except for the hukou and region, the results of other variables are the same. This shows that the research results are reliable.

In addition, we use ordered probit to replace the OLS method for regression and test for robustness. The result is shown in Table 9.

According to Table 9, both the direct and indirect effects of air pollution on well-being are significant after the regression method was replaced. According to Table 2, the model is robust, and the environmental protection tax has a covering effect between well-being and air pollution.

### 5.5. Endogenous Problems

The endogenous problem of the model is solved by using the tool variable method. The use of area means as an instrumental variable has been more widely used in empirical studies [49]. To solve the endogenous problem, we combine the tool variable method with the 2SLS method. The third power of the difference between each sample and the overall average value of the sample is selected as the model tool variable.

The outcomes of the first stage regression are displayed in Table 10’s second column. The results demonstrated a significant positive correlation between instrumental variables (Airpolgap3) and air pollution, rejecting the null hypothesis at the 1% level of significance. Additionally, the first stage’s F-value is 12,773.2, much larger than the usual meaningful critical value of 10, thus indicating that the instrument passes the weak instrument test [50] The outcomes of the second stage regression are displayed in column 3 of Table 10. According to the calculations, keeping other factors constant, for samples subjected to effective interventions with instrumental variables, each unit increase in air pollution will reduce well-being by 0.0252 units. The results of the second stage show that the conclusion that subjective air pollution significantly reduces well-being remains robust. The OLS (−0.0352) and 2SLS (−0.0522) estimates indicate that the OLS estimates are on the high side. Therefore, using OLS can overestimate the impact of subjective air pollution on happiness.

## 6. Conclusions and Recommendations

### 6.1. Conclusions

Using CSS (2019) and China’s statistical yearbook, this paper identifies the relationship and mechanisms between subjective air pollution and happiness which complements the factors influencing well-being from the perspective of air pollution. A total of five aspects are used to conduct heterogeneity tests: income, region, urban-rural location, gender, and age. By altering the research methodology and the core variables, the robustness of the findings is demonstrated. We also tested for possible endogenous problems due to omitted variables using the 2SLS instrumental variables method.

The study finds that subjective air pollution reduces people’s well-being, and is heterogeneous in income, region, urban-rural, gender, and age. Subjective air pollution reduces the well-being of those with low income, in central-west and rural areas, males, and those under 45 years old. At the same time, as an effective means to control air pollution, the environmental protection tax can “neutralize” the decrease in well-being caused by air pollution. Moreover, the data reveal the double dividend of the environmental protection tax: it can govern air pollution and promote green development while simultaneously enhancing people’s well-being.

### 6.2. Policy Recommendations

With the continuous improvement in income levels, sacrificing the environment for development is no longer accepted. People are eager for a better living environment and want a happy and satisfying life. Studying the impact of air pollution on well-being has practical policy implications, especially in the context of “peak carbon and carbon neutrality”. Suggestions include: (1) the economy must expand, but not at the expense of air pollution. To achieve air pollution control and high-quality economic development, new industries and business models should be supported and encouraged. (2) The environmental protection tax can significantly govern air pollution and enhance people’s well-being. However, the environmental protection tax system must be continuously reformed and improved, and the “double dividend” of the tax must be fully utilized. It should also be supplemented with tax incentives for green innovation inputs to stimulate the innovation energy of enterprises, promote their transformation and upgrading, and promote better financial performance. (3) China must apply distinct regional development and environmental protection tax policies. The growth of the eastern, central, and western regions, as well as urban and rural areas, is uneven in terms of factor endowment, geographic advantages, and policy support. To coordinate regional growth, various developments and tax policies must be used based on local circumstances. We need to deepen the economic system reform in the rural areas, resuscitate the rural economy, and close the gap between the urban and rural areas. (4) People’s needs for the environment vary depending on their income, region, urban/rural location, gender, and age. The effects of these factors should be taken into account when formulating environmental protection taxes and environmental governance measures. To achieve Pareto’s improvement of social welfare, China should establish a reasonable ecological compensation mechanism between economically developed regions and economically backward regions, give play to the regulatory role of taxation in income distribution, adjust the excessive income, and narrow the income gap. (5) To compensate for the decline in well-being brought on by air pollution, better compensation or additional subsidies should be offered to those who work outdoors.

### 6.3. Limation and Future Prospects

The data used in this study are from the 2019 CSS and 2019 China Statistical Yearbook, both of which are static cross-sectional data that cannot be compared with historical data, making it impossible to know the degree of change. We looked at income, region, urban/rural location, gender, and age in the heterogeneity test of air pollution. Are variables such as work status, education, religion, etc. heterogeneous? Air pollution and well-being may also be mediated by other variables such as environmental protection inclination, health status, public trust, and class satisfaction which should be examined in future studies.

## Figures and Tables

**Figure 1 ijerph-20-02599-f001:**
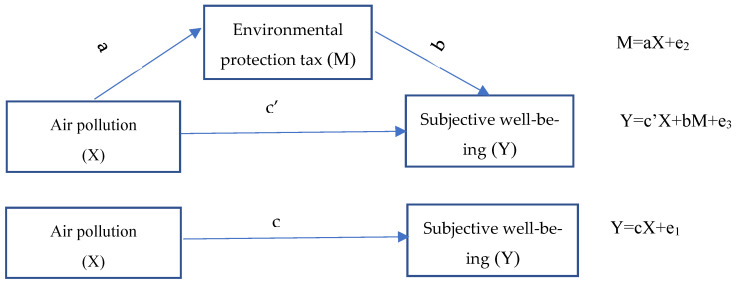
The relationship between subjective air pollution, the environmental protection tax, and well-being. Note: a: The regression coefficient between subjective air pollution and environmental protection tax; b: the regression coefficient between environmental protection tax and well-being; c: the total effects of subjective air pollution on well-being; and c’: the direct effects of subjective air pollution on well-being.

**Figure 2 ijerph-20-02599-f002:**
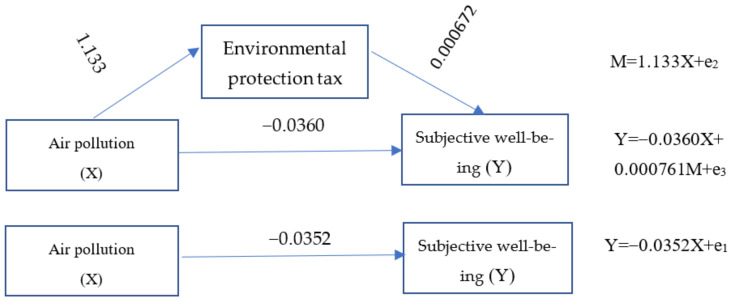
Effect of air pollution on well-being.

**Table 1 ijerph-20-02599-t001:** Descriptive statistical analysis of variables.

Variable		Mean	Std. Dev.	Min	Max
SWB	1 = very unhappy; 2 = not very happy; 3 = relatively happy; 4 = very happy	3.17	0.8	1	4
airpol	1 = not serious; 2 = not very serious; 3 = relatively serious; 4 = very serious	2.06	0.93	1	4
envt	0.73–166.87	64.39	43.3	0.73	166.87
edu	1 = junior high school and below; 2 = high school; 3 = undergraduate; 4 = graduate and above	1.57	0.81	1	4
lninc	0–14.22	8.52	3.43	0	14.22
gender	0 = female; 1 = male	0.43	0.5	0	1
age	18–69 years	46.21	14.21	18	69
age2	324–4761 years	2337.38	1274.74	324	4761
marr	0 = not married; 1 = married	0.8	0.4	0	1
minzu	0 = Minority; 1 = Han	0.92	0.27	0	1
hukou	0 = urban; 1 = rural	0.69	0.46	0	1
work	0 = non-working state; 1 = working state	0.65	0.48	0	1
dzx	1 = east; 2 = central; 3 = west	1.86	0.81	1	3

**Table 2 ijerph-20-02599-t002:** Regression results.

	(1)	(2)	(3)
Variables	SWB	Envt	SWB
airpol	−0.0352 ***	1.133 *	−0.0360 ***
	(0.0124)	(0.657)	(0.0124)
envt			0.000672 **
			(0.000272)
lninc	0.00824 **	−0.130	0.00833 **
	(0.00376)	(0.199)	(0.00375)
edu	0.0453 **	2.742 ***	0.0434 **
	(0.0184)	(0.975)	(0.0184)
gender	−0.0207	−3.061 **	−0.0186
	(0.0240)	(1.268)	(0.0240)
age	−0.0474 ***	−0.496	−0.0470 ***
	(0.00649)	(0.343)	(0.00649)
age2	0.000520 ***	0.00543	0.000517 ***
	(7.00 × 10^−5^)	(0.00370)	(6.99 × 10^−5^)
marr	0.282 ***	4.161 **	0.279 ***
	(0.0347)	(1.836)	(0.0347)
minzu	0.0430	31.08 ***	0.0221
	(0.0438)	(2.319)	(0.0446)
hukou	0.000745	11.69 ***	−0.00711
	(0.0287)	(1.520)	(0.0289)
work	−0.0415	1.760	−0.0427
	(0.0283)	(1.499)	(0.0283)
dzx	−0.0189	−4.092 ***	−0.0162
	(0.0146)	(0.773)	(0.0147)
Constant	3.879 ***	36.86 ***	3.854 ***
	(0.158)	(8.375)	(0.159)
Observations	4.837	4.837	4.837
R-squared	0.025	0.067	0.026

Standard errors in parentheses *** *p* < 0.01, ** *p* < 0.05, * *p* < 0.1.

**Table 3 ijerph-20-02599-t003:** Income heterogeneity.

Variables	(1)	(2)
	High-Income	Low-Income
airpol	−0.0226	−0.0402 ***
	(0.0204)	(0.0156)
Observations	1.600	3.237
R-squared	0.028	0.027
F	4.14	8.23

Standard errors in parentheses *** *p* < 0.01.

**Table 4 ijerph-20-02599-t004:** Region heterogeneity.

	(1)	(2)
Variables	Eastern	Central and Western
airpol	−0.0237	−0.0570 **
	(0.0188)	(0.0222)
Observations	1.993	1.536
R-squared	0.029	0.028
F	6.00	4.45

Standard errors in parentheses ** *p* < 0.05.

**Table 5 ijerph-20-02599-t005:** Discussion on urban/rural heterogeneity.

Variables	(1)	(2)
	Urban	Rural
airpol	−0.0202	−0.0413 ***
	(0.0216)	(0.0151)
Observations	1.492	3.345
R-squared	0.029	0.025
F	4.36	8.67

Standard errors in parentheses *** *p* < 0.01.

**Table 6 ijerph-20-02599-t006:** Discussion on age heterogeneity.

Variables	(1)	(2)
	Elderly	Young
airpol	−0.0318	−0.0372 **
	(0.0226)	(0.0149)
Observations	1.468	3.369
R-squared	0.024	0.028
F	3.28	8.64

Standard errors in parentheses ** *p* < 0.05.

**Table 7 ijerph-20-02599-t007:** Discussion on gender heterogeneity.

Variables	(1)	(2)
	Female	Male
airpol	−0.0187	−0.0592 ***
	(0.0163)	(0.0193)
Constant	3.857 ***	3.876 ***
Observations	2.746	2.091
R-squared	0.023	0.032
F	6.53	6.97

Standard errors in parentheses *** *p* < 0.01.

**Table 8 ijerph-20-02599-t008:** Robustness test.

Variables	(1)	(2)	(3)
	Satis	Envt	Satis
airpol	−0.217 ***	1.169 *	−0.223 ***
	(0.0338)	(0.657)	(0.0337)
envt			0.00433 ***
			(0.000740)
lninc	0.0412 ***	−0.121	0.0417 ***
	(0.0102)	(0.199)	(0.0102)
edu	0.289 ***	2.704 ***	0.277 ***
	(0.0502)	(0.977)	(0.0501)
gender	−0.0931	−3.077 **	−0.0798
	(0.0654)	(1.271)	(0.0652)
age	−0.141 ***	−0.505	−0.139 ***
	(0.0177)	(0.344)	(0.0176)
age2	0.00158 ***	0.00548	0.00156 ***
	(0.000191)	(0.00371)	(0.000190)
marr	0.488 ***	4.354 **	0.469 ***
	(0.0949)	(1.847)	(0.0947)
minzu	0.0313	30.90 ***	−0.102
	(0.120)	(2.325)	(0.121)
hukou	−0.147 *	11.60 ***	−0.197 **
	(0.0783)	(1.524)	(0.0785)
work	−0.0383	1.701	−0.0456
	(0.0772)	(1.502)	(0.0770)
dzx	−0.129 ***	−4.070 ***	−0.112 ***
	(0.0398)	(0.775)	(0.0398)
Constant	9.627 ***	37.05 ***	9.466 ***
	(0.431)	(8.384)	(0.430)
Observations	4.804	4.804	4.804
R-squared	0.047	0.067	0.054

Standard errors in parentheses *** *p* < 0.01, ** *p* < 0.05, * *p* < 0.1.

**Table 9 ijerph-20-02599-t009:** Robustness test.

Variables	(1)	(2)	(3)
	SWB	Envt	SWB
airpol	−0.0470 *** (0.0174)	0.0303 * (0.0158)	−0.0482 *** (0.0174)
envt			0.000928 **
			(0.000384)
	(0.00526)	(0.00479)	(0.00526)
edu	0.0432 *	0.0659 ***	0.0406
	(0.0259)	(0.0235)	(0.0259)
gender	−0.0288	−0.0712 **	−0.0260
	(0.0336)	(0.0305)	(0.0337)
age	−0.0649 ***	−0.0134	−0.0645 ***
	(0.00917)	(0.00827)	(0.00917)
age2	0.000722 ***	0.000152 *	0.000718 ***
	(9.90 × 10^−5^)	(8.91 × 10^−5^)	(9.90 × 10^−5^)
marr	0.362 ***	0.102 **	0.358 ***
	(0.0484)	(0.0442)	(0.0484)
minzu	0.0393	0.865 ***	0.0108
	(0.0614)	(0.0565)	(0.0625)
hukou	0.0103	0.291 ***	−0.000771
	(0.0404)	(0.0367)	(0.0406)
work	−0.0563	0.0704 *	−0.0580
	(0.0399)	(0.0361)	(0.0399)
dzx	−0.0225	−0.133 ***	−0.0188
	(0.0205)	(0.0188)	(0.0206)
Observations	4.837	4.837	4.837
R-squared	0.047	0.067	0.054

Standard errors in parentheses *** *p* < 0.01, ** *p* < 0.05, * *p* < 0.1.

**Table 10 ijerph-20-02599-t010:** Instrumental variable estimation-2SLS.

Variables	First Stage	Second Stage
	Airpol	SWB
Airpolgap3	0.3289045 ***	
	(0.0029)	
Airpol		−0.0252 *
		(0.0146)
Constant	(0.09512)	(0.159)
Observations	4.804	4.804
R2	0.7322	0.0251
F		12773.2

Standard errors in parentheses *** *p* < 0.01, * *p* < 0.1.

## Data Availability

The data is from the following website: http://csqr.cass.cn/index.jsp. The website contains detailed instructions and requirements for obtaining data.

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
