# Peer review of "Air Pollution, Environmental Protection Tax and Well-Being"

_ijerph, 2023, doi:10.3390/ijerph20032599_

Round 1

Reviewer 1 Report

Main findings of this paper; Air pollution reduces the people's well-being. Enviromental protection tax will neutralize the decrease in well-being caused by air pollution.  Environmental tax manage the air pollution, promote green development and enhance people's welbeing.

I see there are some grammatical and writing flaws, should be fixed.

Author Response

Dear reviewer,

      Thank you very much for your valuable comments and suggestions. According to your proposal: “I see there are some grammatical and writing flaws”, we have invited a native English speaker (Ph.D) to help us polish the language of our manuscript.

           Best Regards.

                           Decai Tang

Reviewer 2 Report

The authors present a study on the relationship among air pollution, environmental protection tax and well-being. The topic is relevant and the research work is well organized. I see the potential for this article to be published. Nevertheless, there are some spaces for improvement before publication.

1. In the first section, more background information about the environmental protection tax in China should be provided for those non-Chinese readers. 

2. Similar issue in Section 2.2: there is only one literature from Chinese scholars regarding the connection between air pollution and environmental protection tax. We expect more articles reviewed focusing on China.

3. The authors use SWB for “well-being”. However, in most of the sentences, the word “subjective” (S) is missed.

4. Reference [39] is an introduction about the CSS database but it is in Chinese. I suggest more detail in English in Section 4.1.

5. The coefficients and significance levels of CONSTANT did not show in Table 3~6. The meaning of ***, **, *, as well as the data in brackets did not provide below Tables 2~10.

6. In Section 5.5, more explanations about the IV and test are necessary.

7. Reference: some contains [J] while some has no [J]; the name formats are inconsistent, e.g., X. Zhang in Reference [5] while Zhang, X in Reference [8]. Careful check is required.

Author Response

Dear reviewer, 

    Thank you very much for your valuable comments and suggestions. According to your proposal, we have made a substantial revision of the paper from several aspects, such as background information, reference, explanations about the IV, and so on, so as to get your approval.

  • Point 1: In the first section, more background information about the environmental protection tax in China should be provided for those non-Chinese readers.

Response 1: According to your suggestions, we add more background information about the environmental protection tax in China. Please see Line 52-61.

  • Point 2: Similar issue in Section 2.2: there is only one literature from Chinese scholars regarding the connection between air pollution and environmental protection tax. We expect more articles reviewed focusing on China.

Response 2: There are many papers on environmental protection tax and many papers on well-being, but there are few studies that combine the two, which is the innovation of this paper. We do only find Wang et al., (2022)’ s paper to research the relationship between green tax and happiness with a large-caliber green tax including environmental protection tax.

  • Point 3:The authors use SWB for “well-being”. However, in most of the sentences, the word “subjective” (S) is missed.

Response 3: Direct measurements of happiness or well-being are impossible. Usually, people's subjective feelings are used to gauge it. Typically, when we discuss well-being, we are referring to subjective well-being. Subject well-being is less precise than well-being. Therefore, well-being is used in this paper, but SWB is a widely used shorthand of subjective well-being, so we keep it for readers' benefit.

  • Point 4: Reference [39] is an introduction to the CSS database but it is in Chinese. I suggest more detail in English in Section 4.1.

Response 4: We add some detail about CSS in Section 4.1. Please see Line 222-223.

  • Point 5: The coefficients and significance levels of CONSTANT did not show in Table 3~6. The meaning of ***, **, *, as well as the data in brackets did not provide below Tables 2~10.

Response 5: The heterogeneity tests in Tables 3-6 concentrate primarily on the impact of the core explanatory variables on the dependent variable, ignoring the effects of the control variables and constant terms. We provide the justifications for ***, **, and * that are listed at the bottom of Table 2-10. Please see Line 271,306,324,344,362, 374,394,398,410.

  • Point 6: In Section 5.5, more explanations about the IV and test are necessary.

Response 6: According to your suggestion, we add more explanations about the IV and test. Please see Line 413-421.

  • Point 7: Reference: some contains [J] while some has no [J]; the name formats are inconsistent, e.g., X. Zhang in Reference [5] while Zhang, X in Reference [8]. Careful check is required.

Response 7: Considering your suggestions, we have carefully revised the references in accordance with the format requirements of the journal. Please see the new References from Line Line 593-705.

Reviewer 3 Report

Thank you for giving me this opportunity to read the manuscript entitled "Air Pollution, Environmental Protection Tax and Well-being". The topic of this manuscript is interesting and would be a good contribution to this field. I think it could be considered for publication in IJERPH once the following issues are addressed.

1.     Please replace the keywords that already appear in the manuscript's title with close synonyms or other keywords, which will also facilitate your paper being searched by potential readers.

2.     Line 43- 44, “Air pollution also increases economic and social costs and restricts the sustainable development of economy and society [6].”: a paper titled “Dynamic assessment of PM2. 5 exposure and health risk using remote sensing and geo-spatial big data” is suggested to be added as a reference to support the statement here,

3.     Limitation section should be added as a sub-section to the Discussion.

4.      Some grammatical errors exist in the manuscript. Therefore, a critical review of the manuscript's language will improve its readability.

Author Response

Dear reviewer,

    Thank you very much for your valuable comments and suggestions. According to your proposal, we have made a substantial revision from several aspects such as keywords, more references, and limitations, so as to get your approval.

Point 1: Please replace the keywords that already appear in the manuscript's title with close synonyms or other keywords, which will also facilitate your paper being searched by potential readers.

Response 1: We replace the keywords. Please see Line 31-32. 

Point 2: Line 43- 44, “Air pollution also increases economic and social costs and restricts the sustainable development of economy and society [6].”: a paper titled “Dynamic assessment of PM2. 5 exposure and health risk using remote sensing and geo-spatial big data” is suggested to be added as a reference to support the statement here,

Response 2: We add the paper “Dynamic assessment of PM2. 5 exposure and health risk using remote sensing and geo-spatial big data” as a reference [7]. Please see Line 44.

Point 3: The limitation section should be added as a sub-section to the Discussion.

Response 3: Limitation is discussed in future prospects. We revise the subtitle from future prospects to Limitation and future prospects. Please see Line 472.

Point 4: Some grammatical errors exist in the manuscript. Therefore, a critical review of the manuscript's language will improve its readability.

Response 4: We ask a native English speaker PhD to review manuscript's language to improve its readability.
